# Human *HPSE2* gene transfer ameliorates bladder pathophysiology in a mutant mouse model of urofacial syndrome

Filipa M Lopes[1], Celine Grenier[1], Benjamin W Jarvis[1], Sara Al Mahdy[1], Adrian Lène-McKay[1], Alison M Gurney[2], William G Newman[3,4], Simon N Waddington[5,6], Adrian S Woolf[1], Neil A Roberts[1]*

[1]Division of Cell Matrix Biology and Regenerative Medicine, School of Biological Sciences, Faculty of Biology Medicine and Health, University of Manchester, Manchester, United Kingdom; [2]Division of Pharmacy and Optometry, School of Health Sciences, Faculty of Biology Medicine and Health, University of Manchester, Manchester, United Kingdom; [3]Manchester Centre for Genomic Medicine, Manchester University NHS Foundation Trust, Manchester Academic Health Science Centre, Manchester, United Kingdom; [4]Division of Evolution Infection and Genomics, School of Biological Sciences, Faculty of Biology Medicine and Health, University of Manchester, Manchester, United Kingdom; [5]Maternal & Fetal Medicine, EGA Institute for Women's Health, Faculty of Population Health Sciences, University College London, London, United Kingdom; [6]Wits/SAMRC Antiviral Gene Therapy Research Unit, Faculty of Health Sciences, University of the Witwatersrand, Johannesburg, South Africa

*For correspondence:
Neil.Roberts-2@manchester.ac.uk

**Abstract** Rare early-onset lower urinary tract disorders include defects of functional maturation of the bladder. Current treatments do not target the primary pathobiology of these diseases. Some have a monogenic basis, such as urofacial, or Ochoa, syndrome (UFS). Here, the bladder does not empty fully because of incomplete relaxation of its outflow tract, and subsequent urosepsis can cause kidney failure. UFS is associated with biallelic variants of *HPSE2*, encoding heparanase-2. This protein is detected in pelvic ganglia, autonomic relay stations that innervate the bladder and control voiding. Bladder outflow tracts of *Hpse2* mutant mice display impaired neurogenic relaxation. We hypothesized that *HPSE2* gene transfer soon after birth would ameliorate this defect and explored an adeno-associated viral (*AAV*) vector-based approach. AAV9/*HPSE2,* carrying human *HPSE2* driven by *CAG*, was administered intravenously into neonatal mice. In the third postnatal week, transgene transduction and expression were sought, and ex vivo myography was undertaken to measure bladder function. In mice administered AAV9/*HPSE2*, the viral genome was detected in pelvic ganglia. Human *HPSE2* was expressed and heparanase-2 became detectable in pelvic ganglia of treated mutant mice. On autopsy, wild-type mice had empty bladders, whereas bladders were uniformly distended in mutant mice, a defect ameliorated by AAV9/*HPSE2* treatment. Therapeutically, AAV9/*HPSE2* significantly ameliorated impaired neurogenic relaxation of *Hpse2* mutant bladder outflow tracts. Impaired neurogenic contractility of mutant detrusor smooth muscle was also significantly improved. These results constitute first steps towards curing UFS, a clinically devastating genetic disease featuring a bladder autonomic neuropathy.

## eLife assessment

Urofacial syndrome is a rare early-onset lower urinary tract disorder characterized by variants in HPSE2, the gene encoding heparanase-2. This study provides a **useful** proof-of-principle demonstration that AAV9-based gene therapy for urofacial syndrome is feasible and safe at least over the time frame evaluated, with restoration of HPSE2 expression leading to the re-establishment of evoked contraction and relaxation of bladder and outflow tract tissue, respectively, in organ bath studies. The evidence is, however, still **incomplete**. The work would benefit from the evaluation of additional replicates for several endpoints, quantitative assessment of HPSE2 expression, inclusion of in vivo analyses such as void spot assays or cystometry, single-cell analysis of the urinary tract in mutants versus controls, and addressing concerns regarding the discrepancy in HPSE2 expression between bladder tissue and liver in humans and mice.

## Introduction

Rare early-onset lower urinary tract (REOLUT) disorders comprise not only gross anatomical malformations but also primary defects of functional maturation (*Woolf et al., 2019*). Although individually infrequent, these diseases are together a common cause of kidney failure in children and young adults (*Harambat et al., 2012*; *Woolf, 2022*; *Pepper and Trompeter, 2022*). REOLUT disorders can also have negative impacts on the self-esteem, education, and socialization of affected individuals (*Pepper and Trompeter, 2022*; *Hankinson et al., 2014*). They have diverse phenotypes, including ureter malformations, such as megaureter; bladder malformations, such as exstrophy; and bladder outflow obstruction caused either by anatomical obstruction, as in urethral valves, or functional impairment of voiding without anatomical obstruction. The latter scenario occurs in urofacial, or Ochoa, syndrome (UFS) (*Ochoa, 2004*; *Newman et al., 2013*).

During healthy urinary voiding, the bladder outflow tract, comprising smooth muscle around the section of the urethra nearest the bladder, dilates while detrusor smooth muscle in the bladder body contracts. Conversely, in UFS, voiding is incomplete because of dyssynergia in which the outflow tract fails to fully dilate (*Ochoa, 2004*), and subsequent accumulation of urine in the LUT predisposes to urosepsis (*Ochoa, 2004*; *Osorio et al., 2021*). UFS is an autosomal-recessive disease, and around half of families studied genetically carry biallelic variants in *HPSE2* (*Daly et al., 2010*; *Pang et al., 2010*; *Grenier et al., 2023*). Most are frameshift or stop variants, most likely null alleles, although missense changes, triplication, and deletions have also been reported (*Beaman et al., 2022*). *HPSE2* codes for heparanase-2 (*McKenzie et al., 2000*), also known as Hpa2, which inhibits endoglycosidase activity of the classic heparanase (*Levy-Adam et al., 2010*). Although subject to secretion (*Levy-Adam et al., 2010*; *Beaman et al., 2022*), heparanase-2 has also been detected in the perinuclear membrane (*Margulis et al., 2021*), suggesting yet-to-be defined functions there. The biology of heparanase-2 has been most studied in relation to oncology. For example, in head and neck cancers, heparanase-2 has anti-tumour effects, enhancing epithelial characteristics and attenuating metastasis (*Gross-Cohen et al., 2021*).

The autonomic nervous system controls voiding of the healthy bladder (*Keast et al., 2015*). In fetal humans and mice, heparanase-2 is immunodetected in bladder nerves (*Stuart et al., 2013*; *Stuart et al., 2015*). The protein is also present in pelvic ganglia near the bladder (*Stuart et al., 2015*; *Roberts et al., 2019*). These ganglia contain neural cell bodies that send autonomic effector axons into the bladder body and its outflow tract, with this neural network maturing after birth in rodents (*Keast et al., 2015*; *Roberts et al., 2019*). Mice carrying biallelic gene-trap mutations of *Hpse2* have bladders that fail to fully void despite the absence of anatomical obstruction within the urethral lumen (*Stuart et al., 2015*; *Guo et al., 2015*). Although *Hpse2* mutant mice do have pelvic ganglia, autonomic nerves implicated in voiding are abnormally patterned (*Roberts et al., 2019*). Ex vivo physiology experiments with *Hpse2* mutant juvenile mice demonstrate impaired neurogenic bladder outflow tract relaxation (*Manak et al., 2020*), an observation broadly consistent with the functional bladder outflow obstruction reported in people with UFS (*Ochoa, 2004*). Therefore, while not excluding additional aberrations, such as a central nervous system defect (*Ochoa, 2004*), much evidence points to UFS featuring a genetic autonomic neuropathy affecting the bladder (*Roberts and Woolf, 2020*).

Recently, gene therapy has been used to treat animal models and human patients with previously incurable genetic diseases. Adeno-associated virus 9 (AAV9) has been used as a vector in successful gene therapy for spinal muscular atrophy, an early-onset genetic neural disease (*Mendell et al., 2017*). Here, we hypothesized that human *HPSE2* gene therapy would restore autonomic nerve function in the bladder. Given that UFS is a neural disease, we elected to use *AAV*2/9 (simplified to AAV9) vector, which transduces diverse neural tissues in mice (*Foust et al., 2009*), although uptake into the pelvic ganglia has not been reported. We administered AAV9 vector carrying human *HPSE2* to neonatal mice via the temporal vein, in the first day after birth when bladder neural circuitry is maturing. Several weeks later, evidence of transgene transduction and expression was sought and therapeutic effects on neuropathic dysfunction were investigated in smooth muscle relaxation of the bladder outflow tract and smooth muscle contractility of the bladder body.

## Results

### Administration of AAV9/*HPSE2* to WT mice

We undertook exploratory experiments in WT mice, primarily to determine whether the viral vector was capable of transducing pelvic ganglia after neonatal intravenous injection. We also wished to determine whether administration of AAV9/*HPSE2* was compatible with normal postnatal growth and general health. Given that human gene therapy with AAV9 vectors has been associated with a specific side effect, liver damage (*Mendell et al., 2017*), and the recent implication of AAV2 in community-acquired hepatitis (*Ho et al., 2023*), we also examined mouse livers at the end of the experiment. To determine the efficacy of transduction, neonatal WT mice were administered the AAV9/*HPSE2* vector, and they were followed until they were young adults (*Figure 1a*). Previous studies administering AAV9 carrying reporter genes to neonatal mice have used single doses of up to $10^{12}$ genome copies (*Foust et al., 2009*; *Buckinx et al., 2016*). In the current experiments, because the AAV9/*HPSE2* vector was yet to be tested in vivo, we took a cautious approach, assessing single doses of $2 \times 10^{10}$ or $1 \times 10^{11}$ genome copies. WT neonatal mice administered either dose gained body weight in a similar manner to WT mice injected with vehicle only (*Figure 1b*). Moreover, throughout the 5-week observation period, AAV9/*HPSE2* administered mice displayed normal general appearances and behaviour (e.g. condition of skin and fur, ambulation, grooming, and feeding). As outlined in the 'Introduction', evidence points to UFS being an autonomic neuropathy of the bladder. Therefore, we determined whether the AAV9/*HPSE2* vector had targeted pelvic ganglia that flank the base of the mouse bladder (*Keast et al., 2015*; *Roberts et al., 2019*).

To seek the transduced vector genome, tissue sections were stained using BaseScope in situ hybridization (ISH) for the *WPRE3* genomic sequence (*Figure 1c*, upper panels). The regulatory element was detected in neural cell bodies of ganglia of 5-week-old mice that had been administered either dose of AAV9/*HPSE2*, while no signal was detected in the ganglia of vehicle-only injected mice. Of 42 ganglion cells imaged from three mice injected with the $2 \times 10^{10}$ dose, and 71 ganglion cells imaged from three mice injected with the $10^{11}$ dose, 45% were positive in each group. Here, the definition of positive was at least one red dot in a cell. The transduced cargo was expressed as indicated by the reaction with the human *HPSE2* mRNA probe (*Figure 1c*). No signal was detected in mice that received vehicle-alone. In contrast, *HPSE2* transcripts were detected in the pelvic ganglia of mice that had been administered AAV9/*HPSE2*. Of 44 ganglion cells imaged from three mice injected with the $2 \times 10^{10}$ dose, 11% were positive; and of 57 ganglion cells imaged from three mice injected with the $10^{11}$ dose, 40% were positive. Moreover, although not quantified, the intensity of expression within positive cells appeared greater in the higher dose group (*Figure 1c*).

Liver sections (*Figure 2*) were assessed for *WPRE3* BaseScope signals by counting the number of red dots per unit area. The average density measured in sections from three mice injected with $2 \times 10^{10}$ vector was 587/mm² compared with 835/mm² for three mice injected with $2 \times 10^{11}$ vector, while sections from vehicle-injected mice showed no signal. Assessing *HPSE2* BaseScope signals, the density averaged 45/mm² for three mice injected with $2 \times 10^{10}$ vector, 680/mm² for three mice injected with $2 \times 10^{11}$ vector, and zero for vehicle-injected mice. PSR-stained liver sections, imaged with brightfield and polarized light, are illustrated in *Figure 2b*. The birefringence under polarized light, indicating collagen fibres, was measured as the percentage of the field of view it occupied (*Figure 2c*). There was no significant difference among the three animal groups in the amount of collagen present,

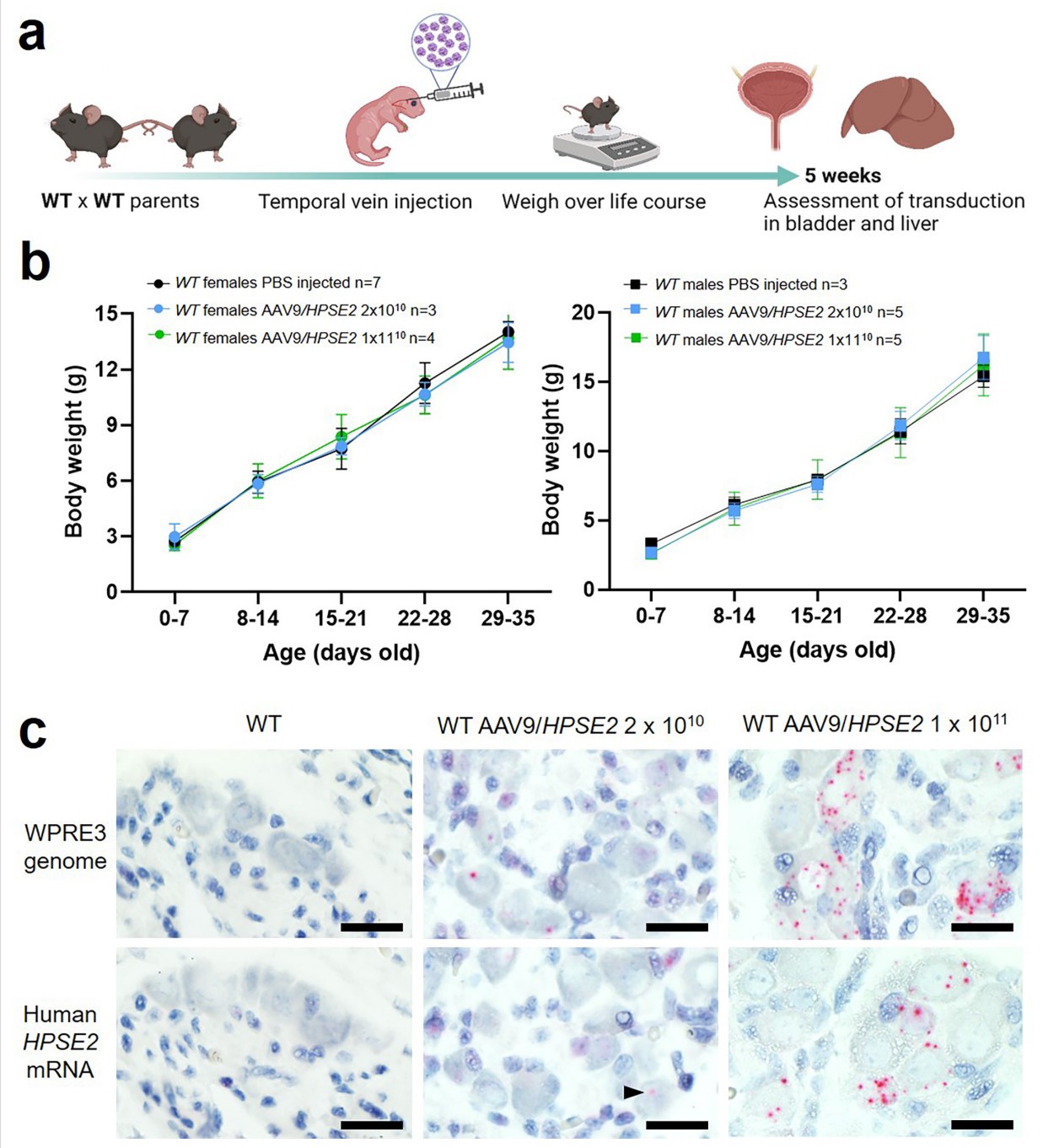

**Figure 1.** Administration of AAV9/*HPSE2* to neonatal WT mice. (**a**) Graphic of study design. The AAV9/*HPSE2* vector genome consisting of flanking AAV2 ITR sequences, the ubiquitous CAG promoter, human *HPSE2* coding sequence, and the WPRE3 sequence. A single dose ($2 \times 10^{10}$ or $1 \times 10^{11}$ genome copies) of AAV9/*HPSE2* was administered to neonates via the temporal vein. Another group of mice received vehicle-only injections. Body weights were monitored, and bladders and livers were harvested for histology analyses at 5 weeks. (**b**) Whole body weights (g). No significant differences were found in growth trajectories comparing $2 \times 10^{10}$ AAV9/*HPSE2*-injected mice with vehicle-only controls (two-way ANOVA); $1 \times 10^{11}$ AAV9/*HPSE2*-injected compared with vehicle-only injected controls; and lower dose compared with higher dose AAV9/*HPSE2*-injected mice. (**c**) BaseScope in situ hybridization (ISH) of the pelvic ganglia. Histology sections are counterstained so that nuclei appear blue. Images are representative of ganglia from three mice in each experimental group. WPRE3 genomic sequence and human *HPSE2* transcripts were not detected in the pelvic ganglia of WT mice injected with vehicle-only (left panel). In contrast, positive signals (red dots) for each probe were detected in the pelvic ganglia of WT mice administered either $2 \times 10^{10}$ (arrow head) or $1 \times 10^{11}$ AAV9/*HPSE2*. For quantification of positive signals, see 'Results'. Bars are 20 µm.

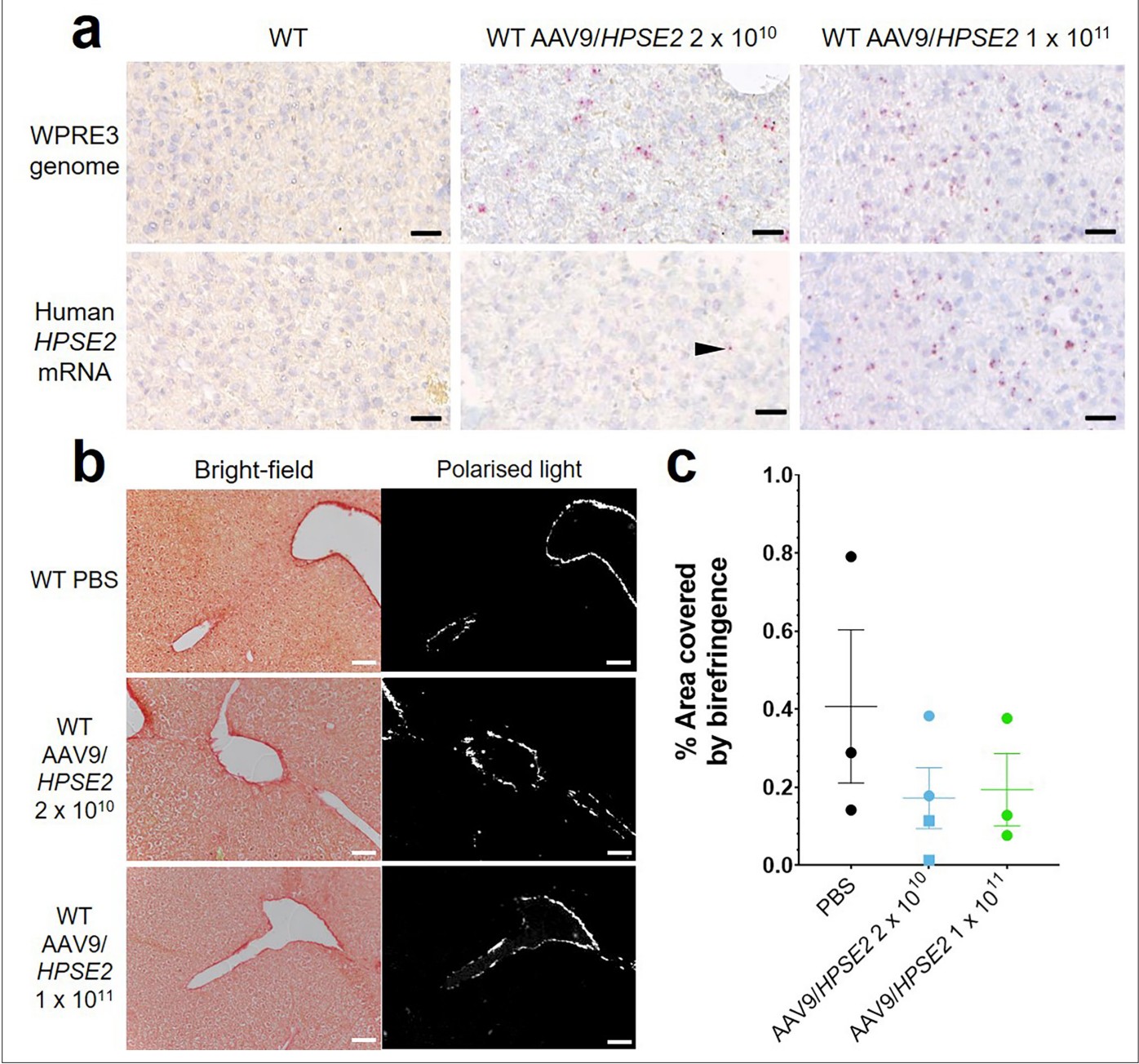

**Figure 2.** Histology of livers of 5-week-old WT mice that had been administered AAV9/*HPSE2* as neonates. In each experimental group, livers from three mice were examined, and representative images are shown. (**a**) BaseScope in situ hybridization (ISH) analyses of livers, with positive signals appearing as red dots. Nuclei were counterstained with haematoxylin. Note the absence of signal for *WPRE3*, part of the AAV9/*HPSE2* genomic cargo, in mice administered PBS only. In contrast, *WPRE3* signal was evident in livers of mice that had received a single dose of $2 \times 10^{10}$ or $1 \times 10^{11}$ genome copies. Regarding human *HPSE2* transcripts, none were detected in the PBS-only livers. Only sparse signals were noted in livers of mice administered the lower AAV dose (arrow head) but the signal for *HPSE2* was prominent in livers of mice administered the higher dose. See the text for quantification of signals. (**b**) PSR staining to seek collagen imaged under direct light (left column) and polarized light (right column). (**c**) There was no significant difference in extent of birefringence between the three groups. Black bars are 30 μm, white are 200 μm.

suggesting that neonatal administration of AAV9/*HPSE2* was not associated with liver fibrosis when assessed in mouse early adulthood.

## Administration of AAV9/*HPSE2* to *Hpse2* Mut mice

Having demonstrated the feasibility and general safety of the procedure in WT mice, a second set of experiments were undertaken to determine whether administration of AAV9/*HPSE2* to Mut mice would ameliorate aspects of their bladder pathophysiology (*Figure 3a*). Reasoning that the higher dose AAV9/*HPSE2* ($1 \times 10^{11}$ genome copies) had been well-tolerated in our scoping experiments in WT mice, we used it here. As expected (*Stuart et al., 2015*), untreated Mut mice gained significantly less body weight than sex-matched WT mice over the observation period (*Figure 3b*). Administration of AAV9/*HPSE2* to Mut mice did not significantly modify their impaired growth trajectory (*Figure 3b*). On autopsy, bladders of untreated Mut mice appeared distended with urine (*Figure 3c and d*), consistent with bladder outflow obstruction, and confirming a previous report (*Stuart et al., 2015*). In contrast, only one of nine bladders of Mut mice that had been administered *AAV9/HPSE2* appeared distended (*Figure 3c and d*). The remainder had autopsy bladder appearances similar to those reported for WT mice (*Stuart et al., 2015*). Bladder bodies were then isolated, drained of urine, and weighed. The bladder body/whole mouse weight (*Figure 3e*) of Mut mice that had not received AAV9/*HPSE2* was significantly higher than untreated WT mice. While values in AAV9/*HPSE2*-administered Mut mice (median 0.0016, range 0.0010–0.0139) tended to be higher than those of untreated WT mice, they were not significantly different from these controls.

Histology sections of the pelvic ganglia were reacted with BaseScope probes (*Figure 4*). Using the *WPRE3* genomic probe, no signals were detected in either WT or Mut mice that had not received the viral vector (*Figure 4a and c*). In contrast, signals for *WPRE3* were detected in WT and Mut mice administered AAV9/*HPSE2* as neonates (*Figure 4b and d*). Using the human *HPSE2* probe, no transcripts were detected in the ganglia of WT and Mut mice (*Figure 4e and g*). In contrast, signals for *HPSE2* were detected in WT and Mut mice administered AAV9/*HPSE2* as neonates (*Figure 4f and h*). In sections from AAV9/*HPSE2*-administered WT mice (n = 3), 63% of 185 ganglion cells were positive for *WPRE3*, and 56% of 210 cells expressed *HPSE2*. In AAV9/*HPSE2*-administered Mut mice (n = 3 assessed), 74% of 167 ganglion cells were positive for *WPRE3*, and 68% of 191 cells expressed *HPSE2*. Heparanase-2 was sought with an antibody reactive to both human and mouse protein (*Figure 4i–l*), with consistent findings in three mice examined in each experimental group. The protein was detected in the pelvic ganglia of untreated WT mice, but only a faint background signal was noted in the ganglia of untreated mutant. Heparanase-2 was detected in the ganglia of both WT and Mut mice that had been administered the vector.

Finally, we determined whether the viral vector transduced cells in the body of the bladder (*Figure 5*). The vector genome sequence *WPRE3* and *HPSE2* transcripts were not detected in the urothelium or lamina propria, the loose tissue directly underneath the urothelium. Within the detrusor muscle layer itself, the large smooth muscle cells were not transduced. However, there were rare small foci of BaseScope signal that may represent nerves coursing through the detrusor.

The vector genome sequence *WPRE3* and *HPSE2* transcripts were also detected in the livers of WT and Mut mice that had been administered the vector (*Figure 6a*). The average number of vector (red dots per area of liver section) from three livers of WT mice injected with AAV9/*HPSE2* were $201/mm^2$ for *WPRE3* and $30/mm^2$ for *HPSE2*. The average values from three Mut mice injected with the AAV9/*HPSE2* were $293/mm^2$ for *WPRE3* and $172/mm^2$ for *HPSE2*. There was no sign of pathological fibrosis in the livers of these mice, as assessed by PSR staining and birefringence quantified under polarized light (*Figure 6b and c*). Kidneys were also analysed using histology but neither the *WPRE3* sequence nor *HPSE2* transcripts were detected in glomeruli or tubules (*Figure 7*).

## *HPSE2* gene transfer ameliorates bladder pathophysiology in *Hpse2* mutant males

We proceeded to undertake ex vivo physiology experiments using myography, with isolated bladder outflow tracts and isolated bladder bodies studied separately. No significant differences in contraction amplitudes induced by 50 mM KCl were documented comparing WT outflow tracts with Mut tissues of mice that had either been administered or had not been administered AAV9/*HPSE2* (*Figure 8a*). Male outflow tracts were pre-contracted with PE, and then subjected to electrical field stimulation

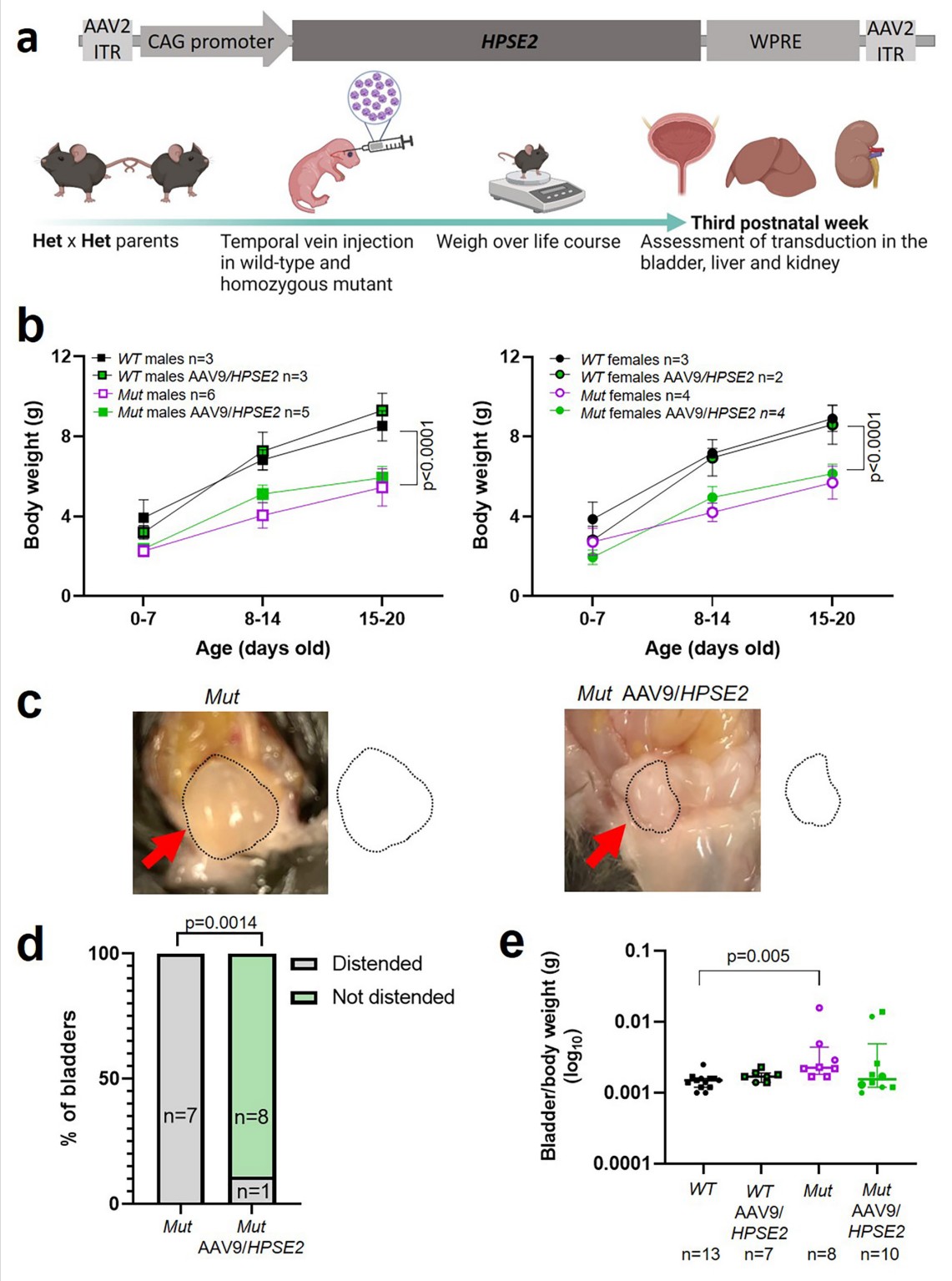

**Figure 3.** Administration of AAV9/*HPSE2* to neonatal mice. (**a**) Graphic of therapy study design, with schematic of AAV9/*HPSE2* vector genome. Heterozygous *Hpse2* parents were mated to generate litters and neonates were genotyped, with WT and Mut offspring used in the study. Some baby mice were not administered the viral vector while others were intravenously administered $1 \times 10^{11}$ AAV9/*HPSE2*. Mice were weighed regularly, and in the third week of postnatal life they were culled and autopsies undertaken to determine whether or not bladders appeared distended with urine. Livers and kidneys were harvested for histology analyses. Bladders were harvested and used either for histology analyses or for ex vivo myography. (**b**) Body weights (g; mean ± SD). Results were analysed with two-way ANOVA. As expected, body growth was impaired in Mut mice that had not received the

*Figure 3 continued on next page*

*Figure 3 continued*
viral vector compared with sex-matched WT mice that did not receive the vector. There was no significant difference in the body growth of Mut mice that either had or had not received AAV9/*HPSE2*. (**c**) Examples at autopsy of a distended bladder in a Mut mouse that had not received the viral vector, and a not-distended bladder in a Mut that had been administered AAV9/*HPSE2* as a neonate. Bladder size indicated by dotted line. (**d**) Untreated Mut mice had distended bladders on autopsy more often than Mut mice that had received AAV9/*HPSE2* (Fisher's exact test). (**e**) Untreated Mut mice had significantly higher empty bladder/whole body weight ratios than untreated WT mice (Kruskal–Wallis test). While viral vector-administered Mut mice tended to have higher empty bladder/whole body weight ratios than WT mice, this was not statistically significant.

(EFS). Nerve-mediated relaxation of pre-contracted outflow tracts had been reported to be impaired in juvenile male Mut mice (*Manak et al., 2020*). Accordingly, as expected, in the current study, neurogenic relaxation of outflow tracts from untreated Mut mice was significantly less than WT relaxation (*Figure 8b and c*). For example, the average Mut relaxation at 15 Hz was around a third of the WT value. In outflow tracts from AAV9/*HPSE2*-administered Mut mice, however, EFS-induced relaxation was no longer significantly different from WT controls, and it was threefold greater than that in untreated Mut mice (*Figure 8b and c*).

Next, male bladder body rings were studied using myography (*Figure 8d–g*). Application of EFS caused detrusor contractions of isolated bladder body rings. In samples from mice that had not received AAV9/*HPSE2* as neonates, contractions were significantly lower in Mut than in WT preparations (*Figure 8d and e*). For example, at 25 Hz the average Mut value was half of the WT value. Strikingly, however, the contractile response to EFS increased around 2.5-fold in Mut bladder rings from AAV9/*HPSE2*-administered mice compared with untreated Mut mouse bladder rings (*Figure 8d and e*). Indeed, EFS-induced DSM contractions in treated Mut mice were not statistically different from those of WT mice (*Figure 8e*). No significant difference in contraction amplitude to 50 mM KCl was documented comparing WT samples with tissues isolated from either vector-administered or untreated Mut mice (*Figure 8f*). As previously reported (*Manak et al., 2020*), bladder body contractions in response to cumulative increasing concentrations of carbachol were significantly higher in untreated Mut tissues compared with WT bladder rings (*Figure 8g*). Bladder body samples from AAV9/*HPSE2*-administered Mut mice showed an attenuated hyper-response to carbachol, and values in this treated Mut group were no longer significantly different from the WT response (*Figure 8g*).

## *HPSE2* gene transfer ameliorates bladder pathophysiology in *Hpse2* mutant females

UFS affects both males and females (*Ochoa, 2004*; *Grenier et al., 2023*), yet results of ex vivo physiology have to date only been reported in tissues from *Hpse2* mutant male mice (*Manak et al., 2020*). Therefore, we proceeded to study outflow tracts and bladder bodies from female Mut mice by ex vivo myography (*Figure 9a–g*). No significant difference in contraction amplitude to 50 mM KCl was documented comparing WT outflow tracts with Mut tissues from mice that had either been administered or not been administered AAV9/*HPSE2* (*Figure 9a*). Female outflow tracts were pre-contracted with vasopressin because they do not respond to PE, the α1 receptor agonist (*Grenier et al., 2023*). In the current study, therefore, vasopressin was used to pre-contract female outflow tracts before applying EFS. Such samples from untreated *Hpse2* female mutant mice showed significantly less relaxation than WT outflow tracts (*Figure 9b and c*), with a striking 20-fold reduction at 15 Hz. Outflow tract preparations from female Mut mice that had been administered AAV9/*HPSE2*, however, displayed EFS-induced relaxation that was not significantly different from WT samples (*Figure 9b and c*). Indeed, the relaxation response of treated Mut samples was significantly increased compared with untreated Mut mice, with an average 12-fold increase in relaxation at 15 Hz (*Figure 9c*).

Next, female bladder body rings were studied with myography (*Figure 9d–g*). Applying EFS caused contractions of isolated bladder body rings (*Figure 9d and e*). In preparations from animals that had not received AAV9/*HPSE2* as neonates, contractions were significantly lower in Mut than in WT preparations, being reduced sevenfold at 25 Hz. Strikingly, however, the contractile detrusor response to EFS in Mut bladder rings from mice that had received AAV9/*HPSE2* was significantly greater than the samples from untreated Mut mice, with a fivefold increase at 25 Hz. However, the bladder body contractions in AAV9/*HPSE2*-administered Mut mice were still significantly smaller than those in WT

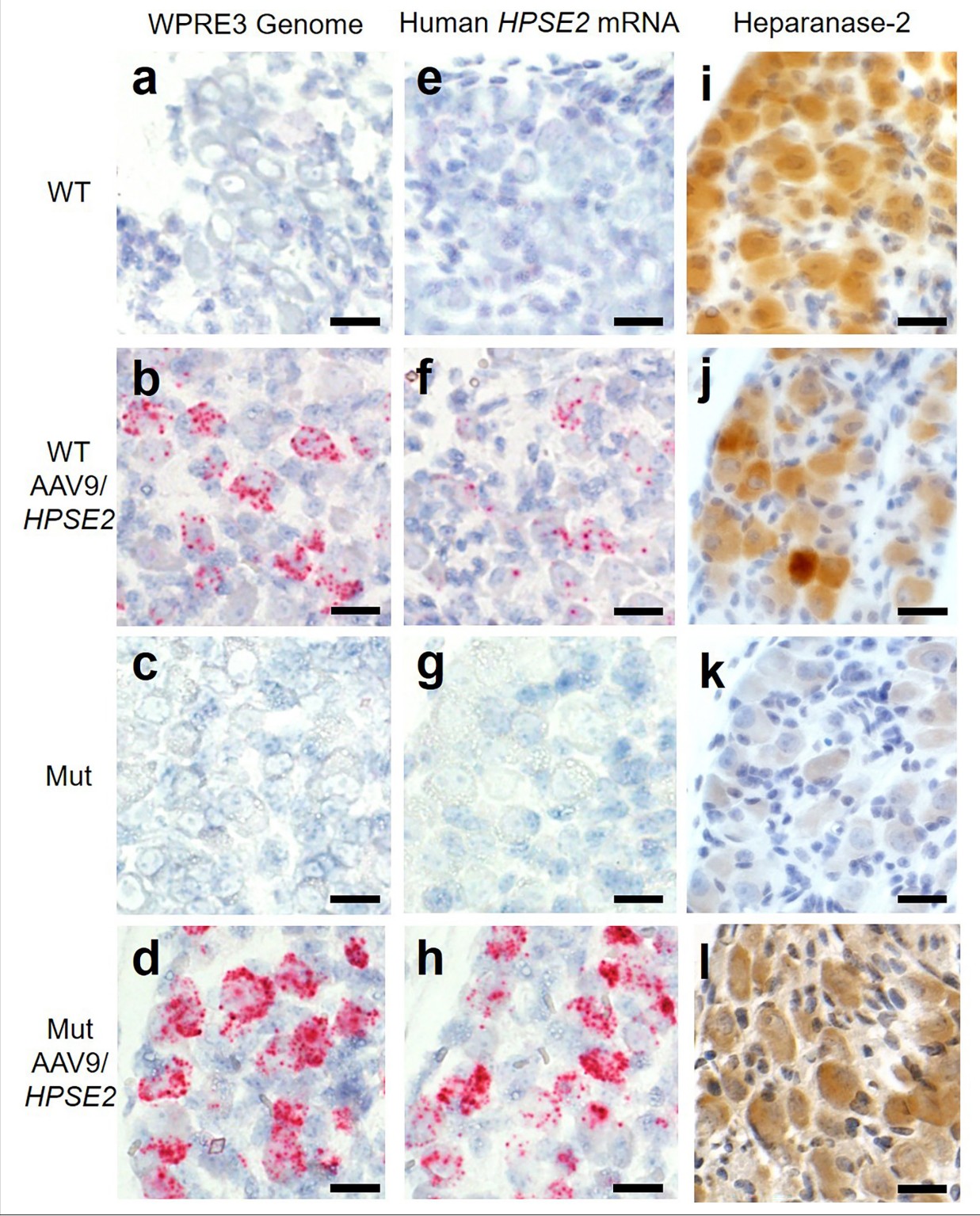

**Figure 4.** Pelvic ganglia histology in the third week of life. BaseScope probes were applied for the *WPRE3* genomic sequence (**a–d**) and for human *HPSE2* transcripts (**e–h**). Other sections were reacted with an antibody to heparanase-2 reactive both human and mouse proteins (**i–l**). The four experimental groups were WT mice that were not administered the viral vector (**a, e, i**); WT mice that had been administered $1 \times 10^{11}$ AAV9/*HPSE2* as neonates (**b, f, j**); Mut mice that were not administered the viral vector (**c, g, k**); and Mut mice that had been administered $1 \times 10^{11}$ AAV9/*HPSE2* as neonates (**d, h, l**). In each group, ganglia from three mice were examined, and representative images are shown. Sections were counterstained with haematoxylin (blue nuclei). Note the absence of BaseScope signals in both WT and Mut mice that had not received the viral vector. In contrast, ganglia

*Figure 4 continued on next page*

*Figure 4 continued*

from WT or Mut mice that were administered AAV9/*HPSE2* displayed signals for both *WPRE3* and *HPSE2*. Note that the signals appeared prominent in the large cell bodies which are postganglionic neurons; signals were rarely noted in the small support cells between the neural cell bodies. See text for quantification of signals. Immunostaining for heparanase-2 showed a positive (brown) signal in all groups apart from ganglia from Mut mice that had not been administered the viral vector; those cells had only a faint background signal. Bars are 20 µm.

mice (*Figure 9e*). No significant difference in contraction amplitude to 50 mM KCl was documented comparing the three bladder body groups (*Figure 9f*). Unlike the enhanced contractile response to carbachol in male Mut bladder rings, detailed above, female bladder rings showed similar responses to carbachol in each of the three experimental groups (*Figure 9g*).

## Discussion

Currently, no treatments exist that target the primary biological disease mechanisms underlying REOLUT disorders. Interventions for UFS are limited and include catheterization to empty the bladder, drugs to modify smooth muscle contractility, and antibiotics for urosepsis (*Ochoa, 2004*; *Grenier et al., 2023*; *Osorio et al., 2021*). Moreover, surgery to refashion the LUT in UFS is either ineffective or even worsens symptoms (*Ochoa, 2004*). Given that some REOLUT disorders have defined monogenic causes, it has been reasoned that gene therapy might be a future option, but the strategy must first be tested on genetic mouse models that mimic aspects of the human diseases (*Lopes et al., 2021*). The results presented in the current study constitute first steps towards curing UFS, a clinically devastating genetic disease featuring a bladder autonomic neuropathy.

We report several novel observations. First, female *Hpse2* mutant mice have neurogenic defects in outflow tract relaxation and detrusor contraction that are similar to those reported for male *Hpse2* mutant mice (*Manak et al., 2020*). This sex equivalence in the mouse model is an important point because UFS affects both males and females (*Grenier et al., 2023*). On the other hand, we observed nuanced differences between the mouse sexes. For example, regarding bladder bodies, females displayed a more profound defect in contractions elicited by EFS, while they lacked the hyper-sensitivity to carbachol shown by males. Second, we have demonstrated that an AAV can be used to transduce pelvic ganglia that are key autonomic neuronal structures in the pathobiology of UFS. Importantly, the neuromuscular circuitry of the bladder matures over the first three postnatal weeks in murine species (*Keast et al., 2015*). A single dose of $1 \times 10^{11}$ genome copies of AAV9/*HPSE2* was sufficient to transduce around half of neural cell bodies in these ganglia when assessed in the third postnatal week. It is possible that the administration of a higher dose of the vector would have resulted in a higher percent of positive cell bodies. Strikingly, heparanase-2 became detectable in the pelvic ganglia of treated Mut mice. Our third key observation is that the $1 \times 10^{11}$ dose significantly ameliorated defect in the mutant LUT despite not every neural cell body being transduced. On autopsy, wild-type (WT) mice had empty bladders, whereas bladders were uniformly distended in mutant mice, a defect ameliorated by AAV9/*HPSE2* treatment. AAV9/*HPSE2* significantly ameliorated the impaired neurogenic relaxation of outflow tracts and the impaired neurogenic contractility of mutant detrusor smooth muscle found in *Hpse2* Mut mice. In future, it will be important to study whether the gene transfer strategy might also ameliorate a dyssynergia between bladder and outflow as assessed by in vivo cystometry (*Ito et al., 2018*). These complex in vivo experiments are, however, beyond the remit of the current work.

Although our data support a neurogenic origin for defects in UFS bladder functionality, we cannot yet rule out an additional myogenic component. In a previous study (*Stuart et al., 2015*), however, we quantified smooth muscle actin (*Acta2*) and myosin heavy chain (*Myh11*) transcripts at 1 and 14 days after birth and recorded no significant differences between WT and *Hpse2* homozygous mutant mice. This suggests that detrusor smooth muscle itself is unchanged in the juvenile period. Moreover, in the current study, we did not detect transduction of AAV9/*HPSE2* into detrusor smooth muscle. A more thorough molecular and structural analysis of the natural history of the detrusor will be an important future work. Similarly, fully elucidating the molecular basis of the UFS neurogenic defect is a key objective. A recent paper has demonstrated the feasibility of single-cell RNA sequencing in murine pelvic ganglia cells (*Sivori et al., 2024*) and so, in future, analysis of WT, *Hpse2* mutant, and *Hpse2*

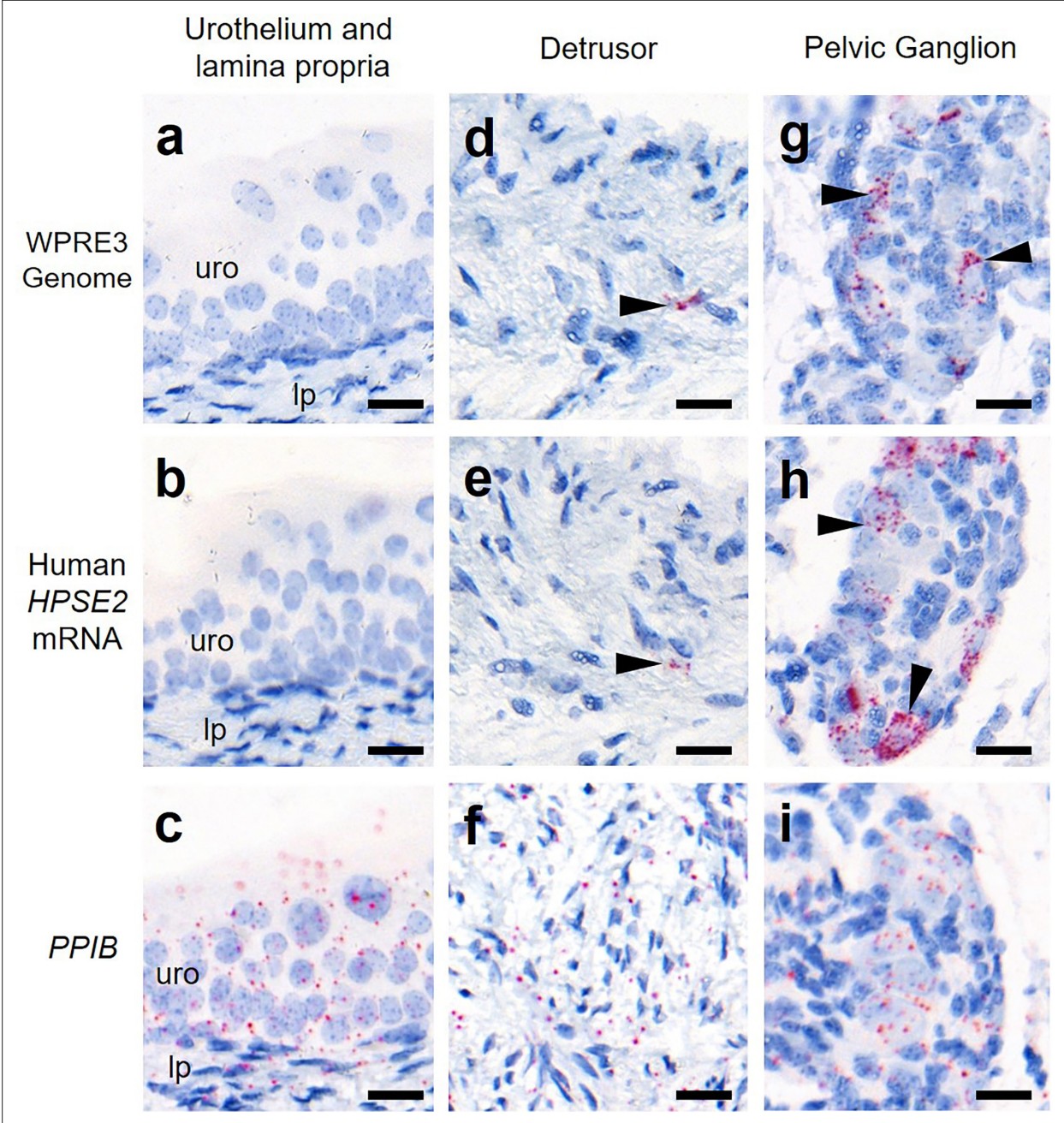

**Figure 5.** Bladder body histology in the third week of life. BaseScope in situ hybridization (ISH) analyses of bladder tissue from AAV9/*HPSE2* administered *Hpse2* mutant mice. Images are representative of three mice examined in this manner. Positive signals appear as red dots and nuclei were counterstained with haematoxylin for the bladder urothelium (uro) and lamina propria (lp) (**a–c**), detrusor smooth muscle layer (**d–f**), and pelvic ganglia body (**g–i**). BaseScope probes were applied for the *WPRE3* genomic sequence (**a, d, g**), human *HPSE2* transcripts (**b, e, h**), and a positive control transcript, *PPIB* (**c, f, i**). Note the absence of BaseScope signals for *WPRE3* and *HPSE2* in the urothelium and lamina propria. There were rare isolated foci of staining in the detrusor layer (indicated by the arrowheads in **d** and **e**) and abundant staining in the pelvic ganglion (arrowheads in **g** and **h**). In contrast, note the widespread expression patterns of the positive control transcript in all tissue types. Scale bars are 20 μm.

mutant-treated pelvic ganglia neurons could reveal the fundamental processes that are aberrant in UFS neurons and are rescued by gene addition treatment.

A feature of the current mouse model, also observed in a different *Hpse2* gene trap mouse (***Guo et al., 2015***), is the poor whole body growth compared with WT controls. This becomes more marked during the first month of life, and, in the current study, it was not ameliorated by neonatal AAV9/*HPSE2* administration. The cause of the growth impairment has not been established, but, as demonstrated

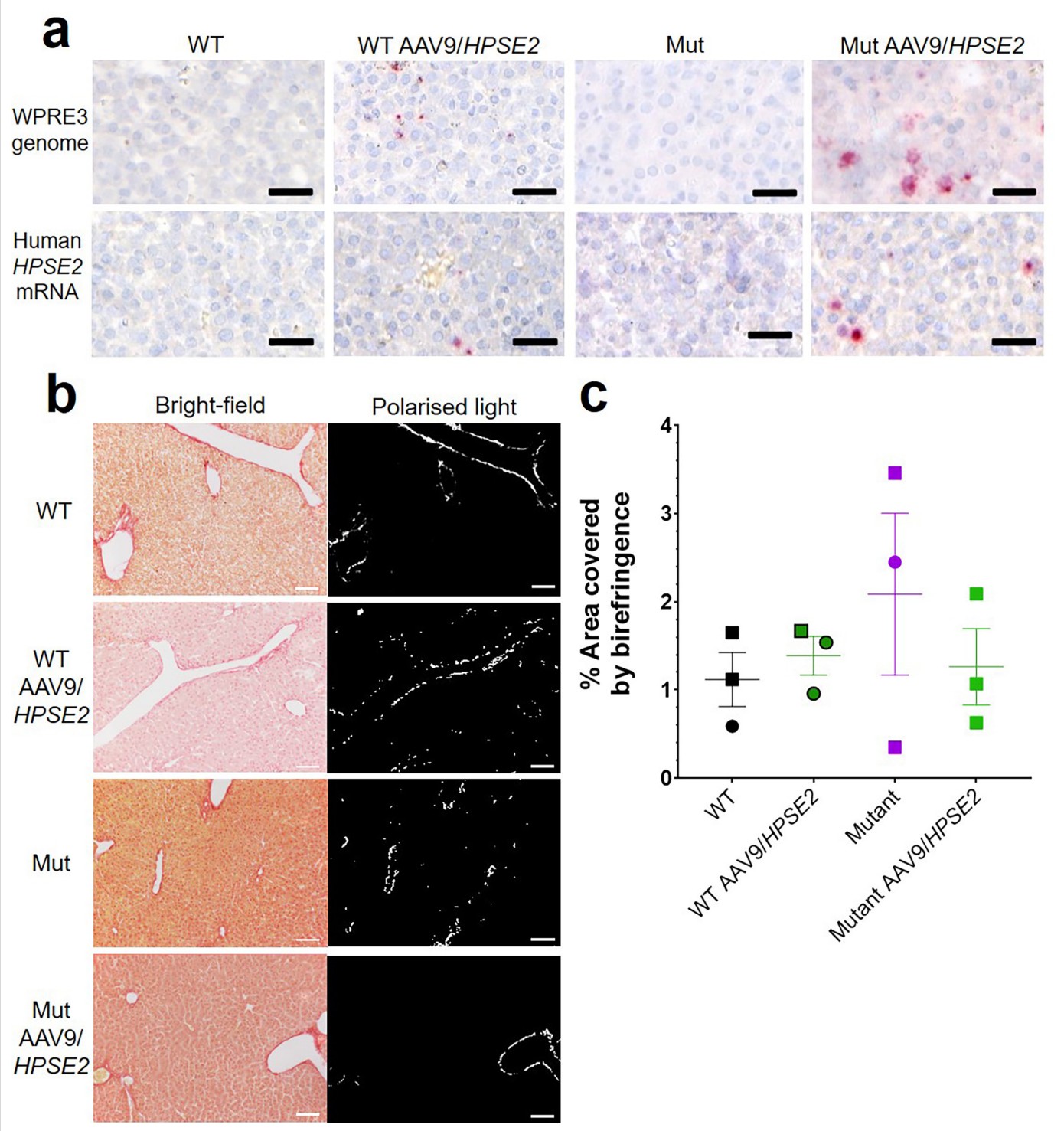

**Figure 6.** Liver histology in the third week of life. The four experimental groups were WT mice that were not administered the viral vector; WT mice that had been administered the viral vector; Mut mice that were not administered the viral vector; and Mut mice that had been administered AAV9/*HPSE2* as neonates. In each group, livers from three mice were examined, and representative images are shown. (**a**) BaseScope in situ hybridization (ISH) analyses of livers, with positive signals appearing as red dots. Nuclei were counterstained with haematoxylin. The vector genome sequence *WPRE3* and *HPSE2* transcripts were detected in the livers of WT and Mut mice that had been administered AAV9/*HPSE2*. See text for quantification. (**b**) PSR staining to seek collagen imaged under direct light (left column) and polarized light (right column). (**c**) There was no significant difference in extent of birefringence between the four groups. Black bars are 30 µm, white are 200 µm.

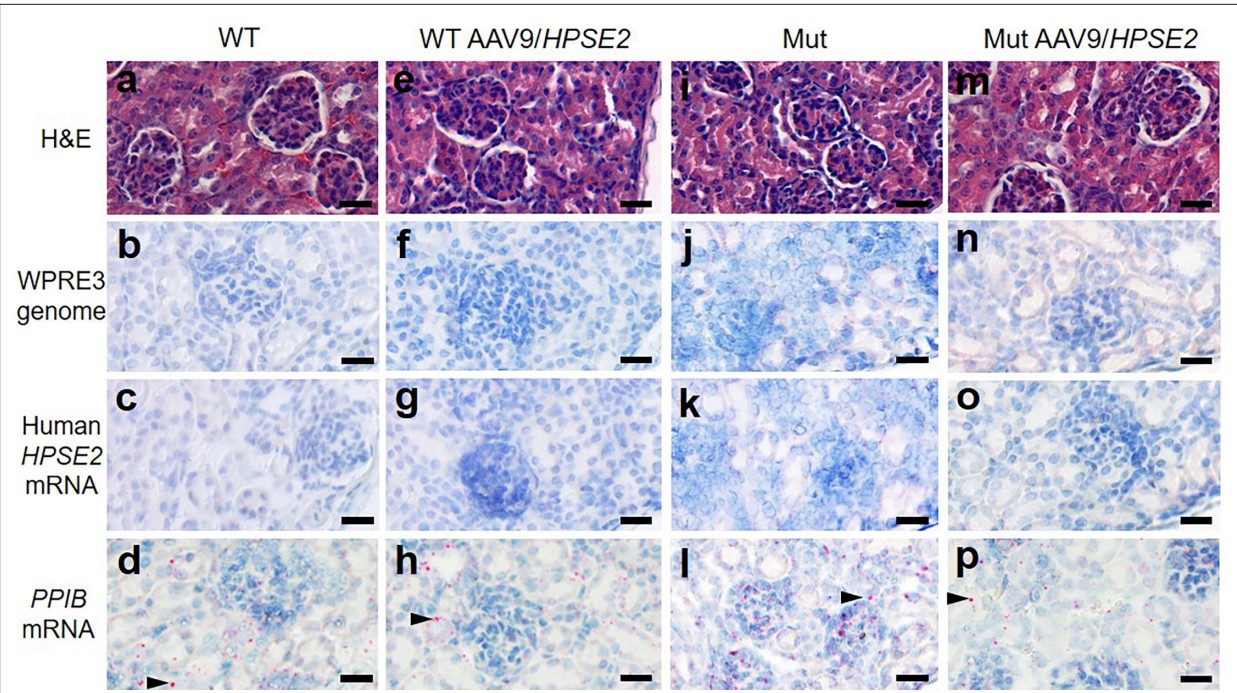

**Figure 7.** Kidney histology in the third week of life. The four experimental groups were WT mice that were not administered the viral vector (**a–d**); WT mice that had been administered 1 × 10¹¹ AAV9/*HPSE2* as neonates (**e–h**); Mut mice that were not administered the viral vector (**i–l**); and Mut mice that had been administered 1 × 10¹¹ AAV9/*HPSE2* as neonates (**m–p**). In each group, kidneys from three mice were examined, and representative images are shown. Some sections were stained with haematoxylin and eosin (**a, e, l, m**) with similar appearances of glomeruli and tubules in all four groups. BaseScope probes were applied for the *WPRE3* genomic sequence (**b, f, j, n**) and for human *HPSE2* transcripts (**c, g, k, o**). Note the absence of BaseScope signals for *WPRE3* and *HPSE2* in all four groups. In contrast, signals (red dots) were noted after application of a BaseScope probe for the house-keeping transcript *PPIB* (**d, h, l, p**) as shown by the arrowheads. Bars are 20 μm.

in the current study, is not accompanied by structural kidney damage. Moreover, our unpublished observations found that *Hpse2* mutant pups appear to feed normally, as assessed by finding their stomachs full of mother's milk. Histological examination of their lungs is normal, excluding aspiration pneumonia. Whatever the cause of the general growth failure, it is notably dissociated from the LUT phenotype, given that AAV9/*HPSE2* administration corrects bladder pathophysiology but not overall growth of the mutant mouse. Of note, a recently published study induced deletion of *Hpse2* in young adult mice (*Kayal et al., 2023*), and this was followed by fatty degeneration of pancreatic acinar cells. Whether the line of *Hpse2* mutant mice used in the current study are born with a similar pancreatic exocrine pathology requires further investigation.

A small subset of individuals with UFS carry biallelic variants of *leucine-rich repeats and immunoglobulin-like domains 2* (*LRIG2*) (*Stuart et al., 2013*; *Grenier et al., 2023*). LRIG2, like heparanase-2, is immunodetected in pelvic ganglia (*Stuart et al., 2013*; *Roberts et al., 2019*). Moreover, homozygous *Lrig2* mutant mice have abnormal patterns of bladder nerves and display ex vivo contractility defects in LUT tissues compatible with neurogenic pathobiology (*Roberts et al., 2019*; *Grenier et al., 2023*). In future, the neonatal gene therapy strategy outlined in the current study could be applied to ameliorate bladder pathophysiology in *Lrig2* mutant mice. As well as in UFS, functional bladder outflow obstruction occurs in some individuals with prune belly syndrome (*Volmar et al., 2001*) and in megacystis microcolon intestinal hypoperistalsis syndrome (*Gosemann and Puri, 2011*). Moreover, apart from UFS, functional bladder outflow obstruction can be inherited as a Mendelian trait and some such individuals carry variants in genes other than *HPSE2* or *LRIG2*. These other genes are implicated in the biology of bladder innervation, neuromuscular transmission, or LUT smooth muscle differentiation (*Weber et al., 2011*; *Caubit et al., 2016*; *Woolf et al., 2019*; *Houweling et al., 2019*; *Mann et al., 2019*; *Beaman et al., 2019*; *Hahn et al., 2022*). Genetic mouse models exist for several of these human diseases and, again, could be used as models for gene therapy.

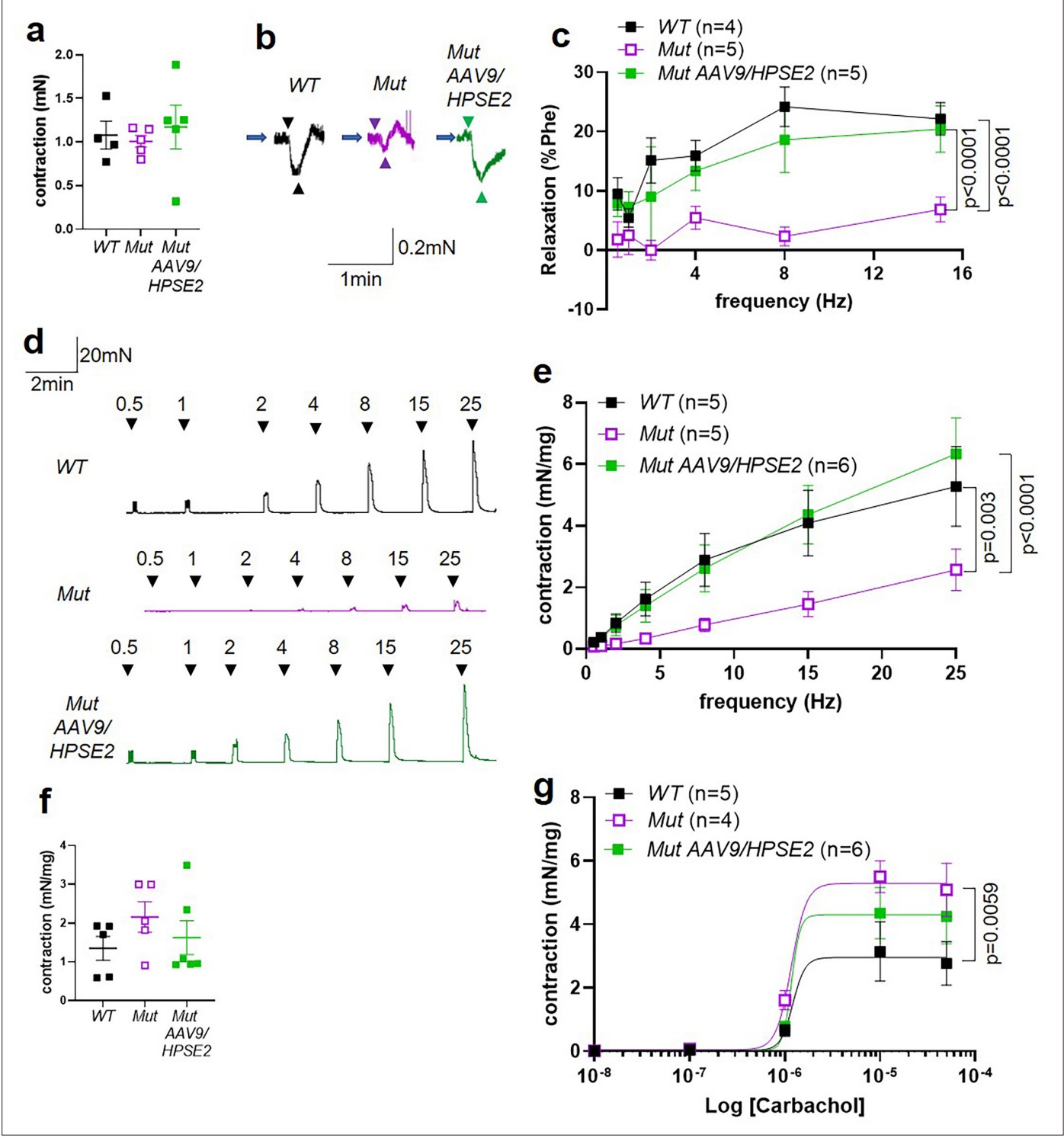

**Figure 8.** Ex vivo myography in males. (**a–c**) are bladder outflow tracts and (**d–g**) are bladder body rings. (**a**) Amplitudes of contraction evoked by 50 mM KCl in male WT (n = 4), Mut (n = 5) and Mut AAV9/*HPSE2* (n = 5) outflow tracts. (**b**) Representative traces of relaxation evoked in WT, Mut, and Mut AAV9/*HPSE2* outflow tracts in response to electrical field stimulation (EFS) at 8 Hz, with arrowheads indicating the start and end of stimulation. (**c**) Relaxations (mean ± SEM) evoked by EFS, plotted as a function of frequency in WT (n = 4), Mut (n = 5), and Mut AAV9/*HPSE2* (n = 5) outflow tracts. (**d**) Representative traces of contractions evoked in WT, Mut, and Mut AAV9/*HPSE2* bladder body rings in response to EFS at the frequencies indicated. (**e**) Amplitude of contractions (mean ± SEM) evoked by EFS in bladders from WT (n = 5), Mut (n = 5), and Mut AAV9/*HPSE2* (n = 6) mice plotted as a function of frequency. (**f**) Amplitudes of contraction evoked by 50 mM KCl in WT (n = 5), Mut (n = 5), and Mut AAV9/*HPSE2* (n = 6) bladders. (**g**) Contraction (mean ± SEM) of bladder rings from WT (n = 5), Mut (n = 4), and Mut AAV9/*HPSE2* (n = 6) mice in response to cumulative application of

*Figure 8 continued on next page*

*Figure 8 continued*

10 nM to 50 µM carbachol, plotted as a function of carbachol concentration. Curves are the best fits of the Hill equation with EC50 = 1.21 µM and Emax = 2.96 mN/mg in WT mice compared with EC50 = 1.20 µM and Emax = 5.30 mN/mg for Mut mice and EC50 = 1.17 µM and Emax = 4.3 mN/mg in Mut AAV9/*HPSE2* mice. comparing WT and Mut by two-way ANOVA with repeated measures.

There are numerous challenges in translating work in mouse models to become effective therapies in humans. First, neonatal mice are immature compared to a newborn person, and the anatomy of their still developing kidneys and LUTs can be compared to those organs in the human foetus in the late second trimester (*Lopes and Woolf, 2023*). Thus, gene replacement therapy for people with UFS and related genetic REOLUT disorders may need to be given before birth. In fact, UFS can present in the fetal period when ultrasonography reveals a dilated LUT (*Grenier et al., 2023*; *Grenier et al., 2023*). The ethics and feasibility of delivering genes and biological therapies to human foetuses are current topics of robust debate (*Sagar et al., 2020*; *Mimoun et al., 2023*) and, with advances in early detection and therapies themselves, may become established in coming years. Second, there are concerns about the possible side effects of administering AAV vectors (*Srivastava, 2023*). Factoring weight for weight, the dose of AAV9 vector administered to the *Hpse2* mutant mice in the current study is of a similar order of magnitude as those used to treat babies with spinal muscular atrophy (*Mendell et al., 2017*). In that report there was evidence of liver damage, as assessed by raised blood transaminases in a subset of treated patients. In the current mouse study, we found that the viral vector transduced livers, while kidneys were resistant to transduction. As assessed by histological analyses to seek fibrosis, livers from AAV9/*HPSE2*-treated mice appeared normal. We cannot, however, exclude transient liver damage, which could be assessed by alterations in blood transaminases. The fact that transduced livers expressed *HPSE2* transcripts, and the observation that heparanse-2 is a secreted protein (*Levy-Adam et al., 2010*; *Beaman et al., 2022*), raises the possibility that the livers of treated mice may act as a factory to produce heparanase-2 that then circulated and had beneficial effects on the LUT, akin to gene therapy to replace coagulation factors missing in haemophilia (*Chowdary et al., 2022*).

In summary, we used a viral vector-mediated gene supplementation approach to ameliorate tissue-level neurophysiological defects in UFS mouse bladders. This advance provides a proof of principle to act as a paradigm for treating other genetic mouse models of REOLUT disorders. In the longer term, application to humans may need to consider fetal therapy for reasons detailed above.

## Methods
### Experimental strategy
In a first set of experiments to explore the feasibility of transducing pelvic ganglia, neonatal homozygous WT mice were intravenously administered the AAV9/*HPSE2* vector (described below). These mice were observed until they were young adults when their internal organs were collected to seek evidence of transgene transduction and expression. Next, we determined whether neonatal administration of AAV9/*HPSE2* to homozygous *Hpse2* mutant mice, hereafter called Mut, would ameliorate ex vivo bladder physiological defects that had been described in juvenile *Hpse2* mutants (*Manak et al., 2020*). The third postnatal week was used as the end point because *Hpse2* Mut mice subsequently fail to thrive (*Stuart et al., 2015*) and ex vivo physiological bladder aberrations have been characterized in Mut mice of a similar age (*Manak et al., 2020*).

### The viral vector
The custom-made (Vector Biolabs) AAV9 vector carried the full-length coding sequence for human *HPSE2* and the broadly active synthetic *CMV early enhancer/chicken Actb* (*CAG*) promoter. This promoter has been used successfully in human gene therapy for spinal muscular atrophy (*Mendell et al., 2017*). The construct also contained the woodchuck hepatitis virus post-transcriptional regulatory element (*WPRE3*), known to create a tertiary structure that increases mRNA stability and thus potentially enhancing biological effects of genes delivered by viral vectors (*Loeb et al., 1999*; *Lee et al., 2005*), and flanking *AAV2* inverted terminal repeat (*ITR*) sequences.

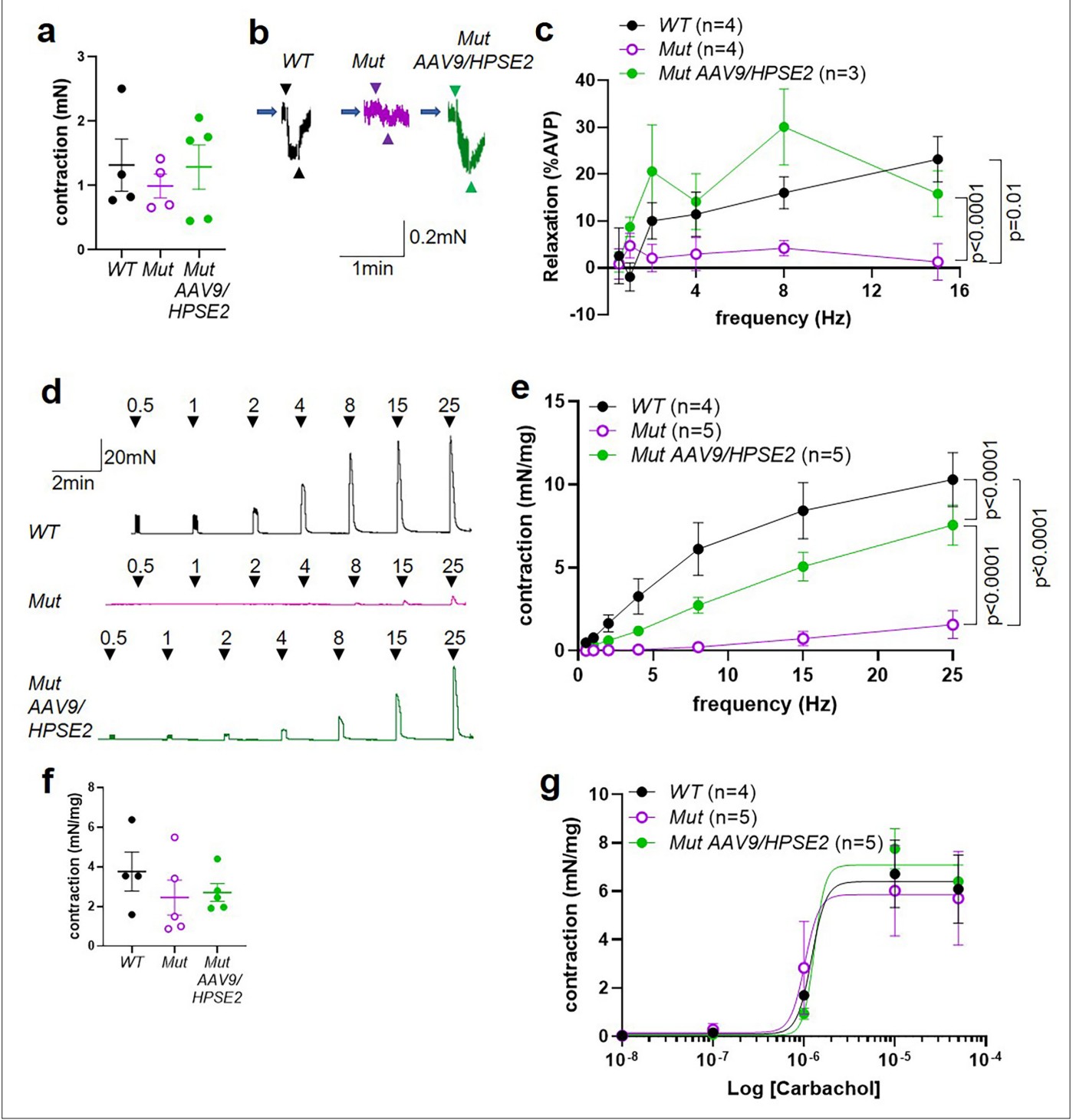

**Figure 9.** Ex vivo myography in females. (**a–c**) are outflow tracts and (**d–g**) are bladder body rings. (**a**) Amplitudes of contraction (mean ± SEM) evoked by 50 mM KCl in female WT (n = 4), Mut (n = 4), and Mut AAV9/*HPSE2* (n = 5) outflow tracts. (**b**) Representative traces of relaxation evoked in female WT, Mut, and Mut AAV9/*HPSE2* vasopressin pre-contracted outflow tracts in response to electrical field stimulation (EFS) at 8 Hz. Arrowheads indicate the start and the end of stimulation. (**c**) Relaxations (mean ± SEM) evoked by EFS, plotted as a function of frequency in female WT (n = 4), Mut (n = 4), and Mut AAV9/*HPSE2* (n = 3) outflows. (**d**) Representative traces of contraction evoked in female WT, Mut, and Mut AAV9/*HPSE2* bladder body rings in response to EFS at the frequencies indicated. (**e**) Amplitude of contraction (mean ± SEM) evoked by EFS in bladder body rings from WT (n = 4), Mut (n = 5), and Mut AAV9/*HPSE2* (n = 5) mice plotted as a function of frequency. (**f**) Amplitudes of contraction evoked by 50 mM KCl in female WT (n = 4), Mut (n = 5), and Mut AAV9/*HPSE2* (n = 5) bladders. (**g**) Contraction of bladder rings (mean ± SEM) from WT (n = 4), Mut (n = 5), and Mut AAV9/*HPSE2* (n = 5) mice in response to cumulative application of 10 nM to 50 µM carbachol, plotted as a function of carbachol concentration. Curves are the best fits of

*Figure 9 continued on next page*

*Figure 9 continued*

the Hill equation with EC50 = 1.22 µM and Emax = 6.40 mN/mg in WT mice compared with EC50 = 1.02 µM and Emax = 5.86 mN/mg for Mut mice and EC50 = 1.30 µM and Emax = 7.08 mN/mg in Mut AAV9/*HPSE2* mice.

## Experimental mice

Mouse studies were performed under UK Home Office project licences PAFFC144F and PP1688221. Experiments were undertaken mindful of ARRIVE animal research guidelines (*Percie du Sert et al., 2020*). C57BL/6 strain mice were maintained in a 12 hr light/dark cycle in the Biological Services Facility of the University of Manchester. The mutant allele has the retroviral gene trap VICTR4829 vector inserted into intron 6 of *Hpse2* (*Stuart et al., 2015*) (mouse accession NM_001081257 and Omnibank clone OST441123). The mutation is predicted to splice exon 6 to a *Neo* cassette and generate a premature stop codon. Accordingly, homozygous mutant mouse bladder tissue contains a truncated *Hpse2* transcript and lacks full-length *Hpse2* transcripts extending beyond the trap (*Stuart et al., 2015*). Mating heterozygous parents leads to the birth of WT, heterozygous, and Mut offspring in the expected Mendelian ratio (*Stuart et al., 2015*). In the first month after birth, Mut mice gain less weight than their WT littermates (*Stuart et al., 2015*) and, because they later fail to thrive, Mut mice are culled around a month of age before they become overtly unwell. Application of tattoo ink paste (Ketchum, Canada) on paws of neonates (i.e. in the first day after birth) was used to identify individual mice, and same-day genotyping was undertaken. This was carried out using a common forward primer (5'CCAGCCCTAATGCAATTACC3') and two reverse primers, one (5'TGAGCACTCACTTAAAAGGAC3') for the WT allele and the other (5'ATAAACCCTCTTGCAGTTGCA3') for the gene trap allele. Neonates underwent general anaesthesia with 4% isoflurane, and AAV9/*HPSE2* suspended in up to 20 µl of sterile phosphate-buffered saline (PBS) was injected into the temporal vein. This procedure took around 2 min, after which the baby mice were returned to their mother. Control neonates received either vehicle-only (PBS) injection or were not injected. At the end of the study, mice underwent Schedule 1 killing by inhalation exposure to a rising concentration of carbon dioxide followed by exsanguination.

## Bladder outflow tract and bladder body myography

Ex vivo myography and its interpretation were undertaken using methodology detailed in previous studies (*Manak et al., 2020*; *Grenier et al., 2023*). Functional smooth muscle defects of both the outflow tract and bladder body were previously characterized in juvenile *Hpse2* Mut mice (*Manak et al., 2020*), so mice of the same age were used here. Outflow tracts, which contained SM but not the external sphincter, were separated from the bladder body by dissection. Intact outflow tracts or full-thickness rings from the mid-portion of bladder bodies, containing detrusor smooth muscle, were mounted on pins in myography chambers (Danish Myo Technology, Hinnerup, Denmark) containing physiological salt solution (PSS) at 37°C. The PSS contained 122 mM NaCl, 5 mM KCl, 10 mM N-2-hydroxyethylpiperazine-N'-2-ethanesulfonic acid (HEPES), 0.5 mM $KH_2PO_4$, 1 mM $MgCl_2$, 5 mM D-glucose, and 1.8 mM $CaCl_2$ adjusted to pH 7.3 with NaOH.

The contractility of outflow tract and bladder body preparations was tested by applying 50 mM KCl to directly depolarize SM and stimulate $Ca^{2+}$ influx through voltage-gated $Ca^{2+}$ channels. Sympathetic stimulation of α-1 adrenergic receptors mediates urethral contraction in male mice and primates but has little effect in females (*Alexandre et al., 2017*). The α-1 adrenergic receptor agonist, phenylephrine (PE, 1 µM), was therefore used to contract male outflow tracts before applying EFS (1 ms pulses of 80 V at 0.5–15 Hz for 10 s) to induce nerve-mediated relaxation (*Burnett et al., 1997*). Female outflow tracts were instead pre-contracted with vasopressin (10 nM) as previously detailed (*Grenier et al., 2023*) because they are known to express contractile arginine-vasopressin (AVP) receptors (*Zeng et al., 2015*). Bladder body rings were subjected to EFS (1 ms pulses of 80 V at 0.5–25 Hz for 10 s) to induce frequency-dependent detrusor contractions. As these contractions are primarily mediated by acetylcholine, through its release from parasympathetic nerves and binding to muscarinic receptors (*Matsui et al., 2002*), we also measured the degree of bladder body contractions evoked directly by the muscarinic agonist, carbachol (10 nM to 50 µM).

## Histology

Samples of bladder body, liver, kidney, and pelvic ganglia flanking the base of the mouse bladder (*Keast et al., 2015*; *Roberts et al., 2019*) were removed immediately after death and prepared for histology. Tissues were fixed in 4% paraformaldehyde, paraffin-embedded, and sectioned at 5 μm for bladders and kidneys, or 6 μm for livers. After dewaxing and rehydrating tissue sections, BaseScope probes (ACDBio, Newark, CA) were applied. ISH was undertaken, using generic methodology as described (*Lopes et al., 2019*). To seek expression of transduced *HPSE2*, BA-Hs-*HPSE2*-3zz-st designed against a sequence in bases 973–1102 in exons 5–7 was used. Other sections were probed for the widely expressed *PPIB* transcript encoding peptidylprolyl isomerase B (BaseScope Positive Control ProbeMouse (Mm)-PPIB-3ZZ 701071). As a negative control, a probe for bacterial *dapB* (BaseScope Negative Control ProbeDapB-3ZZ 701011) was used (not shown). We also used a BaseScope ISH probe to seek genomic *WPRE3* delivered by the viral vector (Cat# 882281). BaseScope detection reagent Kit v2-RED was used following the manufacturer's instructions, with positive signals appearing as red dots within cells. Gill's haematoxylin was used as a counterstain. Images were acquired on a 3D-Histech Pannoramic-250 microscope slide-scanner using a ×*20/0.80 Plan Apochromat* (Zeiss). Snapshots of the slide scans were taken using the Case Viewer software (3D-Histech). Higher-powered images were acquired on an Olympus BX63 upright microscope using a DP80 camera (Olympus) through CellSens Dimension v1.16 software (Olympus). For immunohistochemistry, after dewaxing and rehydration, endogenous peroxidase was quenched with hydrogen peroxide. Slides were microwaved and then cooled at room temperature for 20 min in antigen-retrieval solution (10 mM sodium citrate, pH 6.0). A primary antibody raised in rabbit against heparanase-2 (Abcepta AP12994c; 1:400) was applied. The immunogen was a KLH-conjugated synthetic peptide between 451–480 amino acids in the C-terminal region of human heparanase-2, predicted to cross-react with mouse heparanase-2. The primary antibody was prepared in PBS Triton-X-100 (0.1%) and 3% serum and incubated overnight at 4°C. It was reacted with a second antibody and the positive brown signal generated with DAB peroxidase-based methodology (Vector Laboratories, SK4100). Sections were counterstained with haematoxylin alone, or haematoxylin and eosin. Omission of the primary antibody was used as a negative control. Some liver sections were also reacted with 1% picrosirius red (PSR) solution (diluted in 1.3% picric acid) for 1 hr. Birefringent collagen was visualized under cross-polarized light and the percentage of area covered by birefringence measured as described (*Hindi et al., 2021*).

## Statistics

Data organized in columns are expressed as mean ± SEM and plotted and analysed using the GraphPad Prism 8 software. Shapiro–Wilk test was used to assess normality of data distributions. If the values in two sets of data passed this test and they had the same variance, a Student's *t*-test was used to compare them. If the variance was different, a *t*-test with Welch's correction was used. If data did not pass the normality test, a non-parametric Mann–Whitney test was used. Groups of data with repeated measurements (e.g. at different stimulation frequencies or concentration) were compared using two-way ANOVA with repeated measures. An *F*-test was used to assess the likelihood that independent datasets forming concentration–response relationships were adequately fit by a single curve.

## Acknowledgements

We acknowledge receiving research funding from the Medical Research Council (project grant MR/L002744/1 to ASW and WGN; project grant MR/T016809/1 to ASW, NAR, and FML; and Doctoral Training Programme studentship to BWJ); Kidney Research United Kingdom (project grant Paed_RP/002/20190925 to WGN, GMB, and ASW; and Paed_RP/005/20190925 to NAR and ASW); Newlife Foundation (project grants 15-15/03 and 15-16/06 to WGN and ASW); the Manchester NIHR Biomedical Research Centre (IS-BRC-1215-20007 to WGN); and Kidneys for Life (start-up grant 2018 to NAR, and project grant to NAR 2023); a LifeArc Pathfinder Award (to NAR, ASW, CG, WGN); and an MRC-NIHR UK Rare Disease Research Platform MR/Y008340/1 to NAR, ASW, and WGN. We thank the Manchester NIHR Biomedical Research Centre Rare Diseases theme and the Manchester Rare Condition Centre for support.

# Additional information

## Competing interests

Simon N Waddington: is a co-founder of Bloomsbury Genetic Therapies and is a member of the SMAB of Forge Biologics. The other authors declare that no competing interests exist.

## Funding

| Funder | Grant reference number | Author |
|---|---|---|
| Medical Research Council | MR/L002744/1 | William G Newman<br>Adrian S Woolf |
| Medical Research Council | MR/T016809/1 | Filipa M Lopes<br>Adrian S Woolf<br>Neil A Roberts |
| Medical Research Council | DTP | Benjamin W Jarvis |
| Kidney Research UK | Paed_RP/002/20190925 | William G Newman<br>Adrian S Woolf |
| Kidney Research UK | Paed_RP/005/20190925 | Adrian S Woolf<br>Neil A Roberts |
| Newlife – The Charity for Disabled Children | 15-15/03 | William G Newman<br>Adrian S Woolf |
| Newlife – The Charity for Disabled Children | 15-16/06 | William G Newman<br>Adrian S Woolf |
| Manchester Biomedical Research Centre | IS-BRC-1215-20007 | William G Newman |
| Kidneys for Life | start-up grant 2018 | Neil A Roberts |
| Kidneys for Life | project grant 2023 | Neil A Roberts |
| LifeArc | Pathfinder Award | William G Newman<br>Adrian S Woolf<br>Neil A Roberts |
| Medical Research Council | Rare Disease Research Platform MR/Y008340/1 | William G Newman<br>Adrian S Woolf<br>Neil A Roberts |

The funders had no role in study design, data collection and interpretation, or the decision to submit the work for publication.

## Author contributions

Filipa M Lopes, Investigation, Methodology, Writing – original draft; Celine Grenier, Benjamin W Jarvis, Formal analysis, Investigation; Sara Al Mahdy, Adrian Lène-McKay, Investigation; Alison M Gurney, Methodology; William G Newman, Funding acquisition, Writing – review and editing; Simon N Waddington, Conceptualization, Writing – review and editing; Adrian S Woolf, Conceptualization, Formal analysis, Supervision, Funding acquisition, Investigation, Writing – original draft, Project administration, Writing – review and editing; Neil A Roberts, Conceptualization, Formal analysis, Supervision, Funding acquisition, Investigation, Methodology, Writing – original draft, Project administration, Writing – review and editing

## Author ORCIDs

Adrian S Woolf http://orcid.org/0000-0001-5541-1358
Neil A Roberts http://orcid.org/0000-0002-6955-5536

## Ethics

Mouse experiments were ethically approved by the University of Manchester Biologic Services Facility committee and the UK Home Office (Project Licence PP1688221). Animals were maintained in the Biological Services Facility at The University of Manchester which operates according to the Animals (Scientific Procedures) Act 1986 (ASPA) in conditions that at least meet, and in most cases

exceed, the Codes of Practice for the Care and Welfare of Protected Animals, as laid down by the Home Office.

Reviewer #1 (Public Review): https://doi.org/10.7554/eLife.91828.3.sa1
Reviewer #2 (Public Review): https://doi.org/10.7554/eLife.91828.3.sa2
Author response https://doi.org/10.7554/eLife.91828.3.sa3

## Additional files

### Supplementary files
• MDAR checklist

### Data availability
All data available within the manuscript, note that no large datasets generated or used for this study.

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
