## [Editor Report · eLife assessment]

Urofacial syndrome is a rare early-onset lower urinary tract disorder characterized by variants in HPSE2, the gene encoding heparanase-2. This study provides a **useful** proof-of-principle demonstration that AAV9-based gene therapy for urofacial syndrome is feasible and safe at least over the time frame evaluated, with restoration of HPSE2 expression leading to the re-establishment of evoked contraction and relaxation of bladder and outflow tract tissue, respectively, in organ bath studies. The evidence is, however, still **incomplete**. The work would benefit from the evaluation of additional replicates for several endpoints, quantitative assessment of HPSE2 expression, inclusion of in vivo analyses such as void spot assays or cystometry, single-cell analysis of the urinary tract in mutants versus controls, and addressing concerns regarding the discrepancy in HPSE2 expression between bladder tissue and liver in humans and mice.

---

## [Referee Report · Reviewer #1 (Public Review)]

Summary:

The authors try to use a gene therapy approach to cure urofacial symptoms in an HSPE2 mutant mouse model.

Strengths:

The authors have convincingly shown the expression of AAV9/HSPE2 in pelvic ganglion and liver tissues. They have also shown the defects in urethra relaxation and bladder muscle contraction in response to EFS in mutant mice, which were reversed in treated mice.

Weaknesses:

It is easy to understand that high expression levels of HPSE2 in the bladder tissue lead to bladder dysfunction in human patients, however, the undetectable level of HPSE2 in AAV9 transfected mice bladders is a big question for the functional correction in those HPSE2 mutated mice.

---

## [Referee Report · Reviewer #2 (Public Review)]

In this study, Lopes and colleagues provide evidence to support the potential for gene therapy to restore expression of heparanase-2 (Hpse2) in mice mutant for this gene, as occurs in urofacial syndrome. Building on prior studies describing the nature of urinary tract dysfunction in Hpse2 mutant mice, the authors applied a gene therapy approach to determine whether gene replacement could be achieved, and if so, whether restoration of HPSE2 expression could mitigate the urinary tract dysfunction. Using a viral vector-based strategy, shown to be successful for gene replacement in humans, the authors demonstrated dose-dependent viral transduction of pelvic ganglia and liver in wild type mice. No impact on body weight or liver health was noted suggesting the approach was safe. Administration of AAV9/HPSE2 to Hpse2 mutant mice was associated with similar transduction of pelvic ganglia and a corresponding increase in heparanase-2 protein expression in this site. Analysis of bladder outflow tract and bladder body physiology using organ bath studies showed that re-expression of heparanase-2 in Hpse2 mutant mice was associated with restored neurogenic relaxation of the outflow tract and nerve-evoked contraction of the bladder body, albeit with notable variability in the response at lower frequencies across replicates. Differences were noted in the evoked response to carbachol with bladders from Hpse2 mutant male mice showing increased sensitivity upon HPSE2 replacement compared to wild type, but bladders from female mice showing no difference. Based on these findings the authors concluded that AAV9-based HPSE2 replacement is feasible and safe, mitigates some physiological deficits in outflow tract and bladder tissue from Hpse2 mutant mice and provides proof-of-principle for gene replacement approaches for other genes implicated in lower urinary tract disorders. Strengths include a solid experimental design and data in support of some of the conclusions, and discussion of limitations of the approach. Weaknesses include the variability, albeit acknowledged, in some of the functional assessments, and the limited investigation of bladder tissue morphology in Hpse2 mutant mice.

---

## [Author Response]

The following is the authors’ response to the original reviews.

**Reviewer 1:**
Some important and interesting data are missing. For example, whether the gene therapy can extend the life span of these mutants? The overall in vivo voiding function is missing. AAV9/HSPE2 expression in the bladder wall is not shown.

Our study was not designed to determine whether gene therapy can improve life span of the Hpse2 mutant mice. We know that the mutant mice usually become ill after the first month of life and can die. However, we wanted to study the mice when they were generally well so that there would be no confounding effects on the bladder physiology caused by general ill health. Indeed, a recent study of Hpse2 inducible deletion in adult mice has shown evidence of exocrine pancreatic insufficiency (Kayal et al., PMID 37491420). We are currently exploring the status of the pancreas in our non-conditional juvenile Hpse2 mice, and whether gene transfer into the pancreas is possible.

We strongly agree that in vivo voiding studies will be important in the future, and suggest in vivo cystometry is the gold standard for this but is currently beyond the remit of this study.

It is correct that in this paper we focussed on gene transduction into the pelvic ganglia, because the evidence is mounting that this is a neurogenic disease, with our ex vivo physiological studies showing predominantly neurogenic defects that are corrected by the gene therapy. To further understand the biodistribution of the vector we have now sought evidence of viral transduction into the bladder itself (the new Figure 5). In contrast to the neurons of the pelvic ganglia, we observed very limited transduction: “The vector genome sequence WPRE3, and HPSE2 transcripts, were not detected in the urothelium or lamina propria, the loose tissue directly underneath the urothelium. Within the detrusor muscle layer itself, the large smooth muscle cells were not transduced. However, there were rare small foci of BaseScopeTM signal that may represent nerves coursing through the detrusor.”

**Reviewer 2:**
Weaknesses include a lack of discussion of the basis for differences in carbachol sensitivity in Hpse2 mutant mice, limited discussion of bladder tissue morphology in Hpse2 mutant mice, some questions over the variability of the functional data, and a need for clarification on the presentation of statistical significance of functional data

Yes, it is interesting that untreated male mutant mice have an increased bladder body contraction to carbachol compared with WT males. In a previous paper (Manak et al., 2020) we performed quantitative western blots for the M2 and M3 receptors and found levels were similar in mutants to the WTs, thus the increased sensitivity probably lies post-receptor.

A detailed study of the bladder body is an interesting idea, in terms of possible transgene expression and detailed histology, and is something we will pursue in future studies.

We have reported in our physiology graphs what we find. We do find some variability, particularly at lower frequencies, but our conclusions depend on analyses of the whole curve, which depend on multiple frequencies and show the expected overall pattern of frequency-dependent relaxation.

Thank you, the stats for Figure 8 (now figure 9) have been corrected.

**Reviewer 3:**
Single-cell analysis of mutants versus control bladder, urethra including sphincter. This would be great also for the community.

Yes, in future we are very interested in using a single cell sequencing approach to look at the mutant, WT and rescued pelvic ganglia. In the manuscript we have provided further discussion on the aetiology of urofacial syndrome, and what we still have to learn. We highlight a recent paper in eLife that uses single cell sequencing of mouse pelvic ganglia (Sivori et al., 2024), demonstrating the feasibility of this molecular approach in the pelvic ganglia, and propose this technique could be applied to the study the UFS mice to provide important insights into the molecular pathobiology of the condition.

Detailed tables showing data from each mouse examined.

In theory, it would be very interesting to correlate the strength of human gene transduction into the pelvic ganglia, with, for example, the effect on a physiological parameter. However, in general we used different sets of mice for these techniques so at the present we don’t have this information.

Use of measurements that are done in vivo (spot assay for example). This sounds relatively simple.

We strongly agree that in vivo voiding studies will be important it the future, and suggest in vivo cystometry is the gold standard for this but is currently beyond the remit of this study.

Assessment of viral integration in tissues besides the liver (could be done by QPCR).

This is an important point, and suggest the pancreas is a particularly interesting target for future studies. In the manuscript, we have highlighted a recent study of Hpse2 inducible deletion in young adult mice that has shown evidence of exocrine pancreatic insufficiency (Kayal et al., PMID 37491420), associated with fatty degeneration of pancreatic acinar cells. The Hpse2 mutant animals are smaller than wildtype littermates, the reason for which has not been identified but could be due to defects in processing milk and food. We are currently exploring the status of the pancreas in our non-conditional juvenile Hpse2 mice, and whether gene transfer into the pancreas is possible.

Discuss subtypes of neurons that are present and targeted in the context of mutants and controls.

The make-up of the pelvic ganglia in Hpse2 mutant mice is a fascinating question. Future analysis using scRNA-Seq may be the most effective way to answer this question and is a molecular approach we are looking to pursue in the future.